# How does the process of group singing impact on people affected by cancer? A grounded theory study

Katey Warran,[1] Daisy Fancourt,[2] Theresa Wiseman[3,4]

## ABSTRACT

**Objective** This study aimed to build an understanding of how the process of singing impacts on those who are affected by cancer, including patients, staff, carers and those who have been bereaved.

**Design** A qualitative study, informed by a grounded theory approach.

**Setting and participants** Patients with cancer, staff, carers and bereaved who had participated for a minimum of 6 weeks in one of two choirs for people affected by cancer.

**Methods** 31 participants took part in Focus Group Interviews lasting between 45 min and an hour, and 1 participant had a face-to-face interview.

**Findings** Four overarching themes emerged from the iterative analysis procedure. The overarching themes were: building resilience, social support, psychological dimensions and process issues. Following further analyses, a theoretical model was created to depict how building resilience underpins the findings.

**Conclusion** Group singing may be a suitable intervention for building resilience in those affected by cancer via an interaction between the experience and impact of the choir.

¹School of Social and Political Science, The University of Edinburgh, Edinburgh, UK
²Department of Behavioural Science and Health, University College London, London, UK
³The Royal Marsden NHS Foundation Trust, London, UK
⁴Faculty of Health Sciences, University of Southampton, Southampton, UK

**Correspondence to**
Dr Daisy Fancourt;
d.fancourt@ucl.ac.uk

## Strengths and limitations of this study

► This is the first grounded theory study to have been conducted to explore the impact of group singing for those affected by cancer.
► Thirty-two participants took part spanning patients, staff, carers and bereaved, and saturation was reached.
► This study was concerned with people affected by any type of cancer, but it remains for future studies to establish whether singing has a specific bespoke impact for people with different types of cancer.
► This study used focus group and one-to-one interviews to provide in-depth data to understand shared perspectives and individual experiences, but as no participant observations were undertaken, the behaviours of participants during the singing sessions themselves remain unstudied.

## INTRODUCTION

There is growing awareness about the psychosocial challenges faced by patients with cancer. Major depression rates are approximately five times higher in patients with cancer than the general population, with many cases undiagnosed, and only 30%–35% of patients achieving remission from depression.[1] Patients can also experience more generalised worry, such as heightened sense of vulnerability, inability to make plans and a fear for the future, anger, isolation, diminished self-esteem, concerns over body image, mood disturbance, changes in sexual function and changes in relationships with others.[2 3] These symptoms can be aggravated by changes in patients' daily lives, such as reduced or terminated employment and financial stress. Even 10 years after treatment, 54% of cancer survivors still suffer from at least one psychological issue.[4]

Furthermore, patients with cancer are not the only individuals affected by cancer. Caregivers (whether family or friends) can experience psychosocial challenges commensurate with, or even greater than, those experienced by patients with cancer.[5] Caregivers who suffer bereavement are at increased risk for physical and mental morbidity.[6] Healthcare professionals are also distressed due to emotionally demanding work, resulting in reduced career satisfaction, emotional exhaustion, stress and depression.[7]

In light of these challenges, studies have shown that 1 in 4 female patients with cancer and 1 in 10 male patients with cancer desire psychological support. However, frequently people who most need such support are not those who seek it,[8] and attendance at conventional support groups is low.[9–11]

Although research in this area is still emerging, group singing has been suggested as a suitable intervention for those affected by cancer to support psychosocial needs.[12 13] Just 70 min of singing has been found to be associated with emotional changes, including reductions in self-rated fear, anger, confusion, sadness, tension, tiredness, anxiety and

stress and improvements in energy, happiness, relaxation and social connectedness.[14] Three-month singing interventions for patients with lung cancer have been shown to be associated with improvements in vitality, social functioning and mental health,[12] while 6-month singing interventions have been shown to be associated with improvements in vitality, anxiety and overall mental health for carers and people who have been bereaved and with improvements in anxiety for patients with cancer.[15] Further qualitative studies have identified improved confidence and self-esteem among people affected by cancer who sing in choirs[12] as well as stress relief, friendship and feelings of reward among cancer staff.[16]

However, despite these few promising studies which suggest that group singing is a valuable support, there has been no attempt to build an evidence base regarding *why* singing may be beneficial for those affected by cancer. One previous study proposed that the main mechanism by which singing can affect quality of life in those affected by cancer is that it provides an uplifting musical experience within a supportive community context,[13] but this remains to be explored further. This research gap has been recognised more broadly in the context of other music interventions within health contexts, for example, by DeNora and Ansdell,[17] who state that current research cannot describe *the processes* by which music effects change. Therefore, the aim of this study is to address this gap in the literature and to build an understanding of how the process of group singing impacts on those affected by cancer, including patients, staff, carers and those who have been bereaved.

## METHOD
### Study design
A qualitative methodological approach informed by grounded theory was adopted. This systematic strategy entailed an iterative process of analysing data after each data collection period and constantly comparing data with data. This process allowed a theory of singing and cancer to emerge that was *grounded in the data*[18] as each emergent theme was inductively explored and adapted until the end of data collection when saturation had been reached.[19] Focus group interviews were chosen as the primary method as the research question demanded an understanding of *group* singing and therefore shared perspectives.

### Participants and procedure
This study was part of a larger 2-year non-randomised controlled investigation into singing for people affected by cancer. There were three cohorts recruited: (1) *cohort A:* patients with stage I–III breast or colorectal cancer up to 24 months postdiagnosis, as well as patients with prostate cancer on active surveillance; (2) *cohort B:* hospital staff, people who cared for someone with cancer and those who had lost a family member or somebody they cared for to cancer in the last 3 years and (3) *cohort C:* anybody who had been affected by cancer who wanted to sing in a choir. No musical experience was required to join any of the cohorts, but participants could not also be singing in another choir and join the research. The study was approved by the NHS Research Ethics Service, and participants consented for anonymised data to be used and analysed by the research team.

Participants from all three cohorts were invited to join one of two choirs (based on geographical location) or to be assigned to a 'care as usual' group. Joining the choir involved weekly singing sessions of 60 min preceded by 30 min of socialising and refreshments for a period of up to 24 weeks. The choirs are led by professional choir leaders and the singing typically starts with a short warmup lasting 5–15 min, with the rest of the rehearsal being used to practice contemporary popular music songs. There is no sheet music and no musical experience required, and everyone reads from lyric sheets denoting the choral parts.

For this substudy, participants who had consented to the larger study and joined the choirs were invited to take part. In accordance with grounded theory, the sample selection after the first group was guided by the data collection: the first focus group was conducted with a mixture of patients and non-patients but to explore whether the themes occurred with patients-only and with non-patients only, two further respective focus groups were held. We invited people to take part in the study if they had been to at least six choir rehearsals and were still a member of the choir at the time the study was being conducted (August–October 2017). This totalled 54 eligible individuals who were invited, of whom 32 took part (see table 1 for participant demographics).

### Patient and public involvement
This study was carried out as part of a larger 2-year grant looking at singing for people affected by cancer. Patients with cancer, carers and staff took part in focus groups to design the research questions at the start of the grant and approved all study designs and measures. Patients and public also actively took part in recruitment for the study and have helped to disseminate results from other completed phases of the grant.

### Data collection
Data collection involved three focus group interviews (n=31) held at the site of the choir rehearsals over a 10-week period (August–October 2017) where open questions were asked about the experience of singing in the choir. The first group consisted of patients with cancer and non-patients (staff, carers and bereaved), the second of non-patients only (staff, carers and bereaved) and the third of patients only (current patients and survivors). One participant did not want to contribute to a focus group interview, but did want to participate in the research so had a one-to-one interview. The interviews were all audio-recorded and then transcribed verbatim by a member of the research team.

**Table 1** Demographics of participants who took part in the study, including number of singing sessions attended

| | |
|---|---|
| **Gender** | |
| Female | 21 |
| Male | 11 |
| **Age (mean)** | 63 |
| **Participant group** | |
| Patient | 20 |
| Staff | 2 |
| Carer | 3 |
| Bereaved | 7 |
| **Number of choir sessions attended (mean)** | 29 |
| **Ethnicity** | |
| White | 29 |
| Asian | 2 |
| Prefer not to say | 1 |
| **Education** | |
| AS level/A level | 4 |
| O level/GCSE (General Certificate of Secondary Education) | 6 |
| Vocational training | 3 |
| Undergraduate degree | 9 |
| Postgraduate degree | 10 |

## Data analysis

Data collection and analysis were carried out simultaneously, a key feature of grounded theory methodology. Following guidance from Charmaz,[18 20] this involved carrying out line-by-line initial coding, conceptual focused coding, axial coding to consider relationships between codes, compare categories and subcategories with one another, and theoretical coding to explore and integrate these relationships, as well as creating memos to record emergent ideas. To stay 'close' to the data, analyses was completed by hand as opposed to using software. The coding of the focus groups was conducted independently by two researchers (KW and TW), followed by team meetings and email discussions with all three researchers (DF, KW and TW) to enable collaborative discussion; posing questions and interrogating the data further and refining themes. The interview guide was adapted each time new data were collected to allow for developing ideas to be questioned, and the analysis procedure was repeated each time new data were collected until saturation had been reached. Saturation was agreed between the research team when data were no longer providing new theoretical insights.[20]

## FINDINGS

Four overarching themes and 16 subthemes emerged from the analysis procedure (table 2).

## Building resilience

The participants reported how the choir helped with resilience in relation to their experiences of being affected by cancer. Illustrating this idea, one participant reported:

> …. it helps with things like resilience - because I think you are in a group of people who kinda get it - no matter what… just generally I feel much more open and uplifted… we do sort of light exercises at the beginning, and all of those I think just help to sort of umm strengthen the mental well-being. (Participant from focus group 1)

Resilience and strengthened mental well-being may be created by being with a group of people who have a shared experience, in addition to the physical experience of the choir. Furthermore, the theme of resilience was explored through *coping* (subtheme 1.1):

> It's a wonderful resource when you're feeling a bit stressed… you can actually sort of sing to yourself… it's a very useful skill to have developed to use as a sort of resource of—for managing yourself. (Participant from focus group 1)

The choir was viewed as a medium to bring people together which seemed to support living with cancer, in addition to helping with 'rehabilitation' (participant from focus group 1). The choir was also seen to help coping with specific challenges associated with cancer such as isolation, vulnerability, stress, tiredness, uncertainty and dealing with bereavement.

Furthermore, participants reported that there was something particular about *singing* which supported coping, for example, participants mentioned that there was 'something about the words' which are 'inspiring', 'feeling the harmonies' and 'growing with the music' (participants from focus group 1). As a specific coping tool, participants discussed hearing the songs in their head throughout the week, with one participant describing this as 'a song in my heart' (participant from focus group 3).

It seems then that the songs are a resource for these participants to drawn on throughout the week between rehearsals. Nevertheless, there was one report of an 'earworm' which had negative connotations, where a warmup song was described as an 'awful song' (participant from focus group 3).

Inter-related to the notion of coping, the choirs helped with *building confidence* (subtheme 1.2), including defying negative associations of singing from childhood, giving general confidence and empowering participants at a time where confidence had been lost due to being affected by cancer:

> Participant 1: …at school I was told to mime… I was very sensitive about singing…
>
> Participant 2: I had a similar experience at school… I've joined the choir—it's given me loads of confidence—I can actually sing, I've really surprised myself (Participants from focus group 1)

**Table 2** Description of overarching themes and subthemes, summarising the process and impact of group singing for people affected by cancer

| Theme | Subtheme | Description |
|---|---|---|
| 1. Building resilience | 1.1 Coping | Choir supports coping with cancer and its effects. |
| | 1.2 Building confidence | Choir builds confidence and empowers members. |
| | 1.3 Part of life | Choir has 'lasting effects' beyond the rehearsal itself, providing wider behavioural and social change. |
| 2. Social support | 2.1 Group support (unspoken) | Choir provides a form of 'pastoral', group support that is often 'unspoken'. |
| | 2.2 A fellowship | Choir provides a 'fellowship with other people', and a caring network. |
| | 2.3 Social experience | Choir is a social experience that creates friendship and provides social support. |
| | 2.4 Inclusive | Choir is inclusive and open to all. |
| | 2.5 Organic | Choir grows organically and is 'never static'. |
| 3. Psychological dimensions | 3.1 Emotional | Choir impacts on mood and provides an experience that is 'emotional in a good way'. |
| | 3.2 Positive experience | Choir is an uplifting and positive experience that is fun, enjoyable and a 'nice kind of release'. |
| | 3.3. Holistic | Choir activates the whole body, connecting the psychological and the physical. |
| | 3.4 Identity formation | Choir provides an opportunity to stand there 'as me' and gives 'me time'. |
| 4. Process issues | 4.1 Musical skills | Choir members felt that they attained musical skills via a balance of challenging and fun. |
| | 4.2 Choir leader | The choir leader is someone inspirational who guides, is positive and contributes to improved well-being. |
| | 4.3 Choir logistics | Choir takes place at a good time in the week and the resources are good. |
| | 4.4 Choice of repertoire | There were mixed responses to repertoire with, on the whole, it being viewed as positive and 'uplifting'. |

Just generally giving you more confidence… not just in singing, but in general… that's a benefit. (Interview participant)

It seemed the choir gave confidence to participants in a number of ways, including defying negative associations of singing from childhood, giving general confidence and empowering participants at a time where confidence had been lost due to being affected by cancer.

Moreover, participants reported that the choir was *part of their life* (subtheme 1.3), commenting that the choir provided 'lasting effects' beyond the rehearsal room, prompting wider behavioural change. In relation to confidence, one participant described how she felt more able to give a presentation at work as a consequence of singing in the choir:

My Manager asked me if I would do that [a presentation]… he said 'you don't have to' because he realises that, yeah, that's not the kind of thing I do, but I did say, I would try it, and I think that's partly to do with getting more confidence [from the choir]. (Interview participant)

Other examples included gardening, yoga, making others laugh and singing to grandchildren (participants from focus groups 1 and 3). Thus, it seems that the effects of the choir went beyond the rehearsal room and affected participants' lives throughout the week, including encouraging positive behavioural change.

### Social support

Interconnected to the notion of resilience and a component of it, the choir provided *group support* (subtheme 2.1), where people provided support for one another which was, at times, an unspoken support:

It's almost like an unspoken thing within the group that everybody understands from a certain perspective… that really helps—to feel there are other people that just get it without saying anything… (Participant from focus group 1)

One participant mentioned that it was hard coping with loss of choir members, but she felt supported by the group:

One downside… is losing people… I have taken back strength from the rest of the people here. (Participant from focus group 2)

Another participant felt that some of the lyrics of the music could be emotionally challenging, but the affection and trust of the group helped to provide support:

I think people might struggle with it on an individual level but I think if you're in with a big crowd of people and you can feel the sort of support, yeah and the affection around it—I think much is building in the choir I think that I think that makes it a lot more tolerable… (Participant from focus group 2)

Participants also commented on the empathy that they have for one another (participant from focus groups 2 and 3), in addition to noting that the 'unspoken support' (participant from focus groups 1 and 2) differed from traditional help, such as support groups.

In addition, the choir was seen to provide *a fellowship* with other people (subtheme 2.2):

But it's about the fellowship with the other people… It's actually for everyone. (Participant from focus group 3)

It's just enjoyable and the comradery and the friendship and everything that goes with it. (Participant from focus group 2)

There was a sense that the choristers felt stronger as a consequence of sharing the choir experience; an inclusive place to support one another and to create friendships where everyone is united. Linked to this idea but nuanced from it, participants felt that the choir was also a caring environment:

I haven't been here for a number of weeks, and I had a phone call from Katey [researcher] to find out how I am, and Nina [choir leader], and I thought that was so lovely that, you know, perhaps Polly over there who I thought might worry about where I was, had got somebody to go to say 'you know, have you heard from Lily at all?' I have to say I was touched by it. (Participant from focus group 3)

I've got new friends since I came here. I've now got friends lighting candles for me. I didn't expect that. (Participant from focus group 3)

The participants felt that they had been remembered when they missed rehearsals or were unwell, aiding the formation of a shared fellowship. Inherent in the notion of this fellowship is the *social aspect of the choir experience* (subtheme 2.3). Participants commented on the importance of the first half an hour of the rehearsal, dedicated to socialising:

I like coming and having a cup of tea. I know it sounds really silly little thing, but I've spoken with a friend who is in a local choir and they don't do any of that! (Participant from focus group 3)

Interestingly, the diversity of the friendship group was also commented on. Although choir is a place where members 'all have something in common' (participant from focus group 3), it is also a place where 'you make the most unlikely friends' (participant from focus group 3). Thus, choir brings people together, leading to an *inclusive* experience (subtheme 2.4): welcoming and for people of 'all abilities':

…everybody's welcome—you make us feel very welcome… whoever was in the choir at the beginning and whatever—they've welcomed us—and they've looked as if they're….really pleased to see us! (Participant from focus group 1)

It's nice to have people—all abilities. (Participant from focus group 2)

It was also suggested that due to the commonality of the cancer journey and this shared choir context, people felt equal. One nurse commented on 'seeing everyone come together' (participant from focus group 1), another member of hospital staff said that the choir was 'a good place for me to be' (participant from focus group 1), and another participant said that it was for anyone, including those who 'work in healthcare' (participant from focus group 3). Accordingly, there was a recognition and appreciation among choristers that the choir is open to all.

Moving onto the fifth subtheme in this category which reflects on the dynamics of the group, the *organic* nature of the choir (subtheme 2.5) contributed to a sense of group identity:

Because of the way the choir's forming and developing, the bonding, then it really is a proper sort of a group isn't it? (agreement) even if you've got ebb-ebb and flow. (Participant from focus group 2)

I love it! …it's never static, it's always moving. Just like the music—always moving—like this—and that's what I really like. (Participant from focus group 3)

This group identity is maintained despite the fact that it is ever changing; members come and go, people's lives change and it is constantly developing. There is a dichotomy between the choir as something fleeting 'in the moment' (participant from focus group 1) and the idea that members are 'part of something' as if it had permanence (participant from focus group 3). It is this combination of growing with the choir while feeling a member of it that results in this organic feature to the experience—even though it changes, there is still a sense of group support.

Characteristic of all of the themes in this category and further illustrating this organic quality, there was a moment in the third focus group where members were discussing songs that they would like to sing and they all started singing 'Gaudete' together, a song that they had never learnt as a group. This spontaneity shows that there is a dynamism to the group where they can adapt and work together.

### Psychological dimensions

Participants reported that there were psychological dimensions to the choir experience, for example, they noted that the choir was an *emotional experience* (subtheme 3.1). Several members mentioned this, including that the choir had made them cry. This emotion was described as something 'good':

I found it really emotional too… probably in a good way actually… I think it was a nice kind of release. (Participant from focus group 1)

This emotional experience therefore had a cathartic quality, providing psychological relief whereby participants could 'just express' themselves (participant from focus group 3) and 'let go' (participants from focus groups 1 and 3). Participants were completely immersed in the activity of group singing, resulting in 'a kind of mindfulness' (participant from focus group 1).

Complementing this catharsis, the choir was coined a *positive experience* (subtheme 3.2) with particular emphasis on it being 'uplifting', which was mentioned in all of the focus groups. The positive aspect of the choir was also connected to idea that the choir was fun, enjoyable and a 'happy' experience (participant from focus group 1).

Importantly, these positive emotional experiences were also discussed in relation to the physicality of the choir, emphasising the *holistic* nature of it (subtheme 3.3):

It's physical and like you said about the breathing, its physical and its mental and it's emotional (participant from focus group 1).

It was kind of a mixture of sort of that… need for something that was physical, emotional, and… just all of those things… feel quite holistic to me… (Participant from focus group 1)

Finally, the choir contributed to *identity formation* (subtheme 3.4): while having cancer may have resulted in a loss of identity, such as from not going to work, the choir helped to rebuild life and create a new sense of self:

So where you might have had your work taken away from you, your identity, perhaps stripped a little bit, umm you're not the leader that you know you used to be in the workplace, umm then you know—it gives you something back. (Participant from focus group 1)

Participants also saw the choir as a chance for 'me time' (participant from focus group 2), where the members could have time to just be themselves, rather than be characterised by the experience of living with cancer.

## Process issues

The final overarching theme in this study explores *process issues*, including addressing what the participants sang and how rehearsals were organised. First, the choir supported attainment of *musical skills* (subtheme 4.1):

And you feel so good when you finish a song and it actually sounds alright! (Participant from focus group 2)

These skills were cultivated by the members practising the music at home through use of the CDs provided and through performing in public, which was described as a rewarding and enjoyable experience. Members reported that the choir was the perfect balance of being both challenging, where they were learning something new, and fun:

But it's a combination of the friendliness… as though you're learning something [group agreement] that's what is always so good about it… she's really teaching you—but having a good time as well… (Participant from focus group 2)

As highlighted in this quote, the *choir leader* (subtheme 4.2) was discussed as a key component of the choir:

Our choir leader is I think the key to getting us all together… she's brilliant… I think we've been really lucky. (Participant from focus group 2)

The leadership we've had… from our choir leader… it's a marvel (Participant from focus group 3)

The choir leader was viewed as an anchor, holding everybody together, sustaining the energy of the experience and someone to look up to, and to learn from.

Other aspects of the choir set-up which were viewed as essential components of it being a successful experience included the day of the week and the time of the choir, the building(s) where the choirs took place which were described as 'lovely' and 'like home' (participant from focus group 2), and the consistency of the choir. These *choir logistics* (subtheme 4.3) seemed valuable to participants as they supported regularity in their lives, in contrast to the cancer experience which is often unpredictable:

….literally had 2 weeks off out of 52 of each year… therefore it's providing a form of pastoral care that the hospitals can't do. (Participant from focus group 3)

Finally, the *choice of repertoire* (subtheme 4.4) was discussed as an important component of the choir. Generally, members stated that the repertoire was positive, uplifting, inspirational and that they liked the music:

I've noticed that all of these… are very positive, umm—very…you know—uplifting and the tunes are really you know, again—sort of nice…it's something that makes you feel happy—which I like very much, and I like positive songs. (Participant from focus group 1)

However, there were suggestions that there could be more classical music in one of the focus groups (3), and there were mixed responses to one particularly emotional song that explicitly discussed facing up to cancer.

## A theory for building resilience

Following initial analyses of the data to produce the themes described, the researchers sought to unpack the qualitative mechanisms of the theme 'building resilience', which seemed to underpin all of the findings. A key part of this process was to explore the categories that had been created alongside raw data and researcher memos. Two of the researchers (TW and KW) employed diagramming, comparing and integrating through drawing out thematic maps which prompted an abstract level of analysis.[20] New relationships formed from within the data and memos,

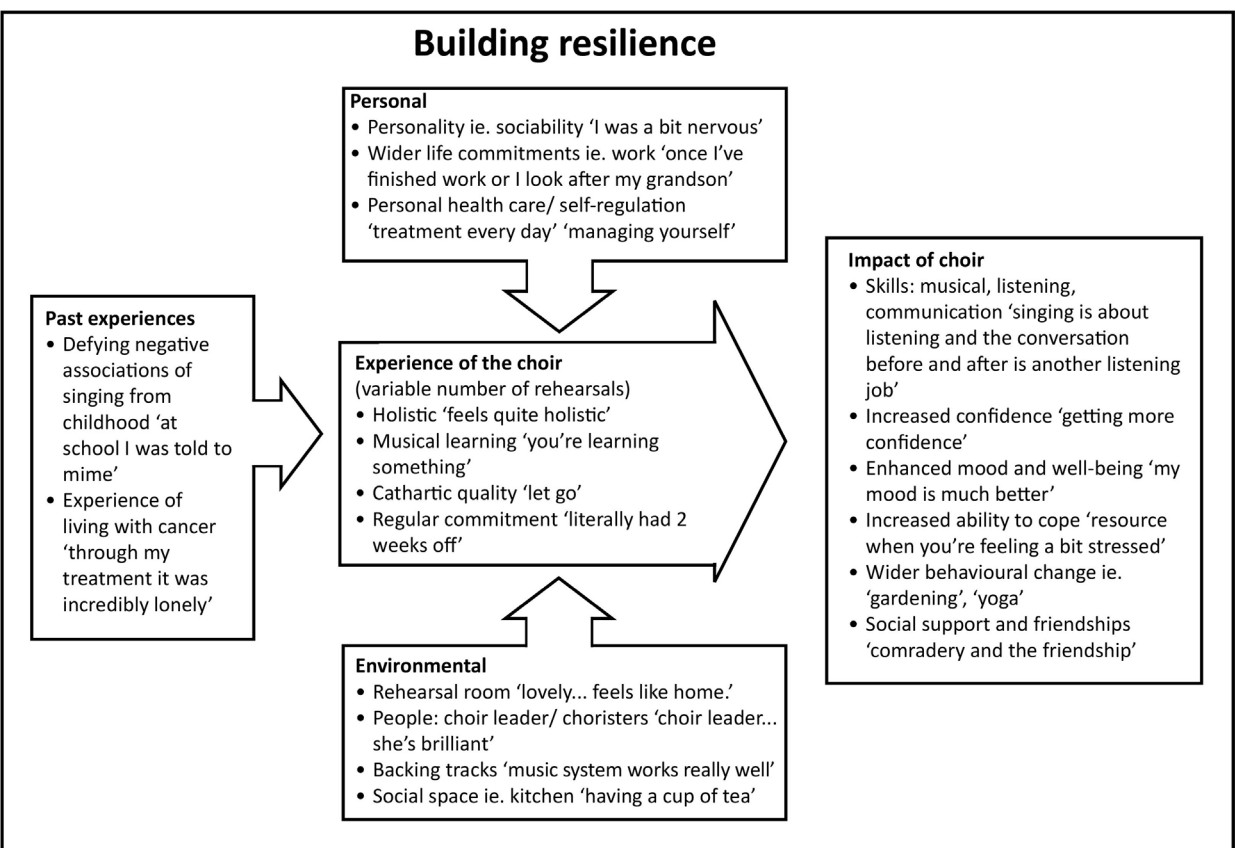

**Figure 1** Theory of building resilience, showing that the perceived resilience created by the choir is underpinned by the interaction of the experience of the choir and the impact of it.

which allowed for reflection on structural elements, such as micro-level (individual) and macro-level (contextual/environmental) features, which were then discussed in light of quotes from the raw data. This interpretative theorising allowed the team to abstract a theory of perceived resilience created out of a dynamic interaction between the experience of the choir—influenced by past experiences of singing from childhood, environment and personal traits—and the impact of it, such as musical skills, confidence and enhanced well-being (figure 1). As a qualitative model grounded in the interconnection of data and theory, it is not possible to stipulate which factors are essential and which are peripheral to building resilience, but this model shows that there are a multiplicity of elements involved and that a combination of them resulted in enhanced, subjective resilience for the participants who took part in this study.

## DISCUSSION

The aim of this study was to build an understanding of how the process of group singing impacts on those affected by cancer. Results revealed four overarching themes (building resilience for those affected by cancer, social support, psychological dimensions and process issues), with further analysis enabling the creation of a model presenting how building resilience underpinned the findings. There were no noticeable differences in themes across the different focus groups, suggesting that these findings are relevant for patients, staff, carers and bereaved. This is also the first known study to provide a model for resilience, highlighting the mechanisms by which singing impacts on those affected by cancer.

The key finding of this study is that group singing appears to contribute to building resilience in those affected by cancer. A general definition of resilience is the capacity to maintain healthy psychological and physical well-being despite being exposed to adversity.[21 22] In light of the psychosocial challenges that can be experienced during cancer, promoting resilience has been described as a critical element of the psychosocial care for patients with cancer, with suggestions that medical staff should recognise and promote mechanisms of adaption.[23] Within cancer care, sense of confidence, self-transcendence of the cancer experience and self-esteem have been proposed as key areas to focus on in order to build resilience.[24]

With this understanding of resilience in mind, the findings of this study support group singing as an important intervention to build resilience in those affected by cancer. Group singing was seen to improve self-confidence, transcend the cancer experience through immersion in the activity and improve self-esteem through acquisition of musical skills via a balance of challenge and fun. One of the key reasons that the choir could achieve

this was also through its specifically *social* nature. Community support can be harnessed for resilience, assisting in developing individual psychological and community strengths.[22 25] The choir made social connections through a group context where a fellowship among members was created. All of these elements are shown in the suggested model for building resilience.

The findings of this study also reinforce other theories concerning psychosocial needs. Lutgendorf *et al*[26] put forward a three-factor model to suggest that social support, emotional expression and benefit finding can aid those affected by cancer, including supporting strong immune response. Choristers expressed emotions in a 'good way', attained social support through new friendships and an inclusive environment, and they found some form of benefit from their experiences through learning new skills, enhanced confidence and the enjoyable experience of singing together. Consequently, the present study suggests that these three factors can contribute to positive adjustment, including promoting resilience.

There are also indications that the results of this study resonate with theories of well-being. The results from this study suggest that the choir might be able to fulfil the three basic psychological needs of relatedness, competence and autonomy.[27] First, the participants commented on the group and social aspects of the choir. The choir also gave a fulfilling experience—offering a sense of achievement by learning songs together—which could satisfy the need for competence. This is also supported by previous studies.[12 15] Members were also autonomous as they chose to come to choir each week, integrated the activity into their lives and were self-regulated in their attitude to learning.

More generally, this study links to concepts of enhanced well-being. In addition to supporting the components of increased quality of life in previous studies,[12 15] it also correlates with notions of hedonic and eudaimonic well-being.[28] Happiness and satisfaction can be seen in themes relating to positive experiences, and psychological functioning and self-realisation are inherent in building resilience and psychological dimensions. This supports previous research which has shown that instrumental learning in adulthood can provide both immediate (hedonic) and long-term (eudaimonic) enhancements to subjective well-being,[29] suggesting that there might be shared mechanisms of music across different musical experiences.

The cathartic and holistic nature of the choir experience also correlates with theories of psychological 'flow',[30] providing activities that are 'oceanic', 'deeply moving' and 'mystical'. These experiences also allow complete focus on the present moment and a feeling of being in control of the environment.[31] Singing in the choir was described as cathartic by participants, in addition to being likened to mindfulness which suggests that it enabled members to focus on the present; a finding also reported in other singing studies.[32 33] The 'mystical' component of flow may be inherent in the emotional and indescribable features of the choir, likening it to a spiritual or transcendent experience. This suggestion is reinforced by other singing studies.[34–36] Moreover, it is interesting that the choristers admired and trusted their choir leader in a way that resounds with spiritual leadership where a communion is created between the leader and their fellows, whether this be choristers or a congregation.[37]

As a qualitative methodology pursuing emergent analytic goals, it is not possible to make generalisations based on this study; however, the insights gained may support the development of future interventions.[18] By breaking down the elements of how choir singing may support building resilience, it may be possible for future quantitative studies to test which of these features form the constitutive elements of resilience and which are incidental. Attaining this understanding may allow for optimisation of benefits from choral singing, understanding better the relationship between singing and resilience for those affected by cancer. Another future avenue could be to conduct an ethnography to understand the contribution of the sociocultural context to this theory of building resilience. A limitation of this study is that it has not been able to reflect on shared patterns of behaviours and values which would enable further reflection on whether cultural components are related to health outcomes.

## CONCLUSION

There is a need to identify interventions which will provide psychosocial support for those affected by cancer, including patients, staff, carers and bereaved. This study has shown that group singing may be a suitable intervention, building resilience in those affected via an interaction between the holistic experience and impact of the choir context.

**Correction notice** Since this paper was first published online the open access licence has been changed from CC-BY-NC to CC-BY.

**Acknowledgements** The authors wish to thank Tenovus Cancer Care, the Royal Marsden NHS Foundation Trust and the choir members who took part in the study, in addition to Saoirse Finn who supported transcription. The authors also wish to acknowledge The Royal Marsden Cancer Charity who funded TW's time and the infrastructure support of the Royal Marsden/Institute of Cancer Research Biomedical Research Centre.

**Contributors** The research team collaboratively designed the study. TW and DF led on the supervision of the project. KW led on the data collection with support and guidance from DF and TW. TW and KW conducted independent analysis which was discussed at team meetings, resulting in the creation of the model presented, drawn by KW. KW produced the draft of the report which was refined following detailed input from TW and DF.

**Funding** This research was funded by Tenovus Cancer Care with additional support from the Arts and Humanities Research Council [AH/P005888/1].

**Competing interests** None declared.

**Patient consent** Not required.

**Ethics approval** The NHS Research Ethics Service approved the study.

**Provenance and peer review** Not commissioned; externally peer reviewed.

**Data sharing statement** The data used in this research were collected subject to the informed consent of the participants. Unfortunately, the dataset is not publicly available as participants only consented to the research team having access to the raw dataset.

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
