## [Reviewer comments · BMJ Open]

This paper was submitted to a another journal from BMJ but declined for publication following peer review. The authors addressed the reviewers' comments and submitted the revised paper to BMJ Open. The paper was subsequently accepted for publication at BMJ Open.

(This paper received three reviews from its previous journal but only two reviewers agreed to published their review.)

ARTICLE DETAILS

TITLE (PROVISIONAL)	How does the process of group singing impact upon people affected by cancer? A grounded theory study
AUTHORS	Warran, Katey; Fancourt, Daisy; Wiseman, Theresa

VERSION 1 – REVIEW

REVIEWER	Kathy Helzlsouer, MD, MHS National Cancer Institute, U.S.A.
REVIEW RETURNED	30-May-2018

GENERAL COMMENTS	The manuscript presents the results of a qualitative study on the experience of group singing by cancer patients and caregivers. The methods were informed by grounded theory. This paper adds to a growing body of literature on the role and benefits of group singing, in this case, for cancer patients, health care providers, and caregivers. This is a well conducted qualitative study based in a larger 2-year investigation with only minor issues to clarify and address in the manuscript. Introduction: lines 42-43. Clarify that cases “of depression” is undiagnosed and that achieving remission applies to depression. Study Design: A table of the baseline population characteristics from which the participants of the qualitative study were drawn from should be included. The larger study had 3 cohorts recruited – presumably these were separate choirs but that is not clear. This figures in later as in the process description of whether there were any comments about the composition of the choirs – Cohort C had a mixed group – how as that perceived? Any differences from the other two cohorts? How many of the enrolled patients had active disease? For the larger study it was noted that participants were invited to join a choir or “assigned” to a usual care group – was this self-selected or was the assignment random? Please describe the eligibility for the larger group and the assignment process. Was there information on prior singing experience that may have informed the discussion?
--

	The methods state that 54 individuals were eligible – please state the eligibility requirements. Data analysis – Were there analytic programs used to code/assess the transcribed data? Please describe how saturation was determined. Findings: Was there anything that came out of the specific focus groups rather than the mixed focus group that is worth noting with respect to either process or the discussion? Building resilience: the carry-over effects noted could be grouped together – it is noted on several different comments (lines 19, 28, 47) Dynamics of the group: A better title here may be social support. Features of learning: A more appropriate subheading may be Process Issues – which includes what they sang, how it was organized. Comment on line 21 – seems more relevant to social support – coping with loss Line 31: did the groups have overlapping participants – (“participant from focus group 2 and 3; participant from group 1 and 2 – or did separate participants from those groups make similar comments?) The only comments specific to dynamics were on line 38 and 41. Only two comments specifically noted dynamic change...”always moving, and ebb and flow. The majority relate to process issues (day of the week, timing, music choice etc.) – and are beneficial to point out for future research. A theory for building resilience: the figure notes past experiences (childhood, music, living with cancer) but these were not brought out in the discussion. Was this information collected? Was it used to inform the interpretation of the comments? Did it come out in the focus groups? The discussion: Would be helpful to note if there were any insights gained on the process that may be helpful for future interventions.
--	--

REVIEWER	Minjung Shim Stony Brook University, United States
REVIEW RETURNED	23-Jun-2018

GENERAL COMMENTS	This study aimed to address an important research question (i.e. identifying the process of resilience building through group singing in people affected by cancer) through a grounded theory study. Reviewing of relevant literature and description of the rationale for the study were done satisfactorily and the study resulted in a relatively strong theoretical model. However, unfortunately the overall quality of the paper was diminished by several limitations. First, there was insufficient description on the grounded theory data analysis process (e.g., axial
---

	coding, selective coding, theoretical coding, memoing etc.). Second, there was a lack of clarity in describing and categorizing the sub-themes. There seemed to be overlap between some of the sub-themes and the description of some sub-themes need more clarification or differentiation from the other themes. The connection between some of the inferences and the supporting quotes by participants was weak. Third, the description of the final grounded theory is also weak. Fourth, some of the inferences made in the discussion and conclusions sections are not justified by the results. Lastly, limitations of the study were not sufficiently addressed in the body of the paper. Thus, I regret to inform the authors that the manuscript cannot be accepted with the current condition.
--	--

VERSION 1 – AUTHOR RESPONSE

Reviewer: 1

Reviewer Name: Kathy Helzlsouer, MD, MHS

Institution and Country: National Cancer Institute, U.S.A.

Please state any competing interests or state 'None declared': None declared

Please leave your comments for the authors below

The manuscript presents the results of a qualitative study on the experience of group singing by cancer patients and caregivers. The methods were informed by grounded theory. This paper adds to a growing body of literature on the role and benefits of group singing, in this case, for cancer patients, health care providers, and caregivers. This is a well conducted qualitative study based in a larger 2-year investigation with only minor issues to clarify and address in the manuscript.

We'd like to thank Dr Helzlsouer for her comments and are pleased that she sees this as a well conducted study that adds to the literature on the benefits of singing. Many thanks also for the detailed feedback that follows in relation to the minor issues mentioned.

Introduction: lines 42-43. Clarify that cases "of depression" is undiagnosed and that achieving remission applies to depression.

We'd like to thank Dr Helzlsouer for highlighting the need to clarify here. We have now indicated that the remission applies to depression, as per the original source paper.

Study Design:

A table of the baseline population characteristics from which the participants of the qualitative study were drawn from should be included.

We also collected data of participants' ethnicity and education, and this has now been included in our demographics table.

The larger study had 3 cohorts recruited – presumably these were separate choirs but that is not clear. This figures in later as in the process description of whether there were any comments about the composition of the choirs – Cohort C had a mixed group – how as that perceived? Any differences from the other two cohorts?

We apologise for the lack of clarity here. The participants from all three cohorts were invited to join one of two choirs. This has now been updated in our manuscript.

How many of the enrolled patients had active disease?

We apologise that unfortunately we do not have access to this information. When participants enrolled into our larger quantitative study, we asked one question regarding whether they had cancer or previously had it. It is therefore not possible for us to distinguish between participants who had cancer at the time of the focus groups from those who previously had it.

For the larger study it was noted that participants were invited to join a choir or “assigned” to a usual care group – was this self-selected or was the assignment random? Please describe the eligibility for the larger group and the assignment process. Was there information on prior singing experience that may have informed the discussion?

We again apologise for the lack of clarity here. We have now updated the manuscript to explain that the larger study was a non-randomised controlled trial, mentioning that no musical experience was needed to join.

The methods state that 54 individuals were eligible – please state the eligibility requirements.

We have now explained that people who had previously enrolled to join the larger study were eligible to join this substudy if they had been to at least six choir rehearsals and were still a member of the choir at the time the study was being conducted (August-October 2017).

Data analysis – Were there analytic programs used to code/assess the transcribed data? Please describe how saturation was determined.

We have now explained that we did our analyses without a software programme and that saturation was agreed between the research team when data was no longer providing new theoretical insights.

Findings:

Was there anything that came out of the specific focus groups rather than the mixed focus group that is worth noting with respect to either process or the discussion?

We thank Dr Helzlsouer for this question; however, we found that the themes from each group were interrelated and complemented one another. We have now noted in the discussion that there were no noticeable differences in themes across the different focus groups.

Building resilience: the carry-over effects noted could be grouped together – it is noted on several different comments (lines 19, 28, 47)

We thank Dr Helzlsouer for this suggestion. As the second reviewer commented that there was a lack of clarity in our describing and categorising of sub-themes, we have chosen to show our process more clearly by labelling these elements as distinct, but interrelated, sub-themes. We hope this is sufficient and also addresses this comment.

Dynamics of the group: A better title here may be social support.

We thank Dr Helzlsouer for this suggestion and agree that it is a better title. We have now updated this.

Features of learning: A more appropriate subheading may be Process Issues – which includes what they sang, how it was organized.

We again thank Dr Helzlsouer for this suggestion and have now updated our manuscript.

Comment on line 21 – seems more relevant to social support – coping with loss

We have now moved the comments about coping with loss into the section on social support.

Line 31: did the groups have overlapping participants – (“participant from focus group 2 and 3; participant from group 1 and 2 – or did separate participants from those groups make similar comments?)

We apologise for the lack of clarity here – these were separate participants from those groups who made similar comments. We have now updated this to the plural ‘participants’ which should solve this.

The only comments specific to dynamics were on line 38 and 41.
Only two comments specifically noted dynamic change...”always moving, and ebb and flow.

As recommended, we have updated the title of this section to ‘social support’.

The majority relate to process issues (day of the week, timing, music choice etc.) – and are beneficial to point out for future research.

As recommended, we have updated the title of this section to ‘process issues’.

A theory for building resilience: the figure notes past experiences (childhood, music, living with cancer) but these were not brought out in the discussion. Was this information collected? Was it used to inform the interpretation of the comments? Did it come out in the focus groups?

We thank Dr Helzlsouer for highlighting this. All of this information came from the focus groups. We have now included additional quotes to show this in both the main body of our findings and in our model, in addition to explaining in greater detail how the model was created.

The discussion:

Would be helpful to note if there were any insights gained on the process that may be helpful for future interventions.

We thank Dr Helzlsouer for this recommendation. We have now included a paragraph explaining our insights which would be helpful for future interventions. We have reflected on how our work may improve future quantitative studies to optimise benefits in relation to building resilience.

Reviewer: 2

Reviewer Name: Minjung Shim

Institution and Country: Stony Brook University, United States

Please state any competing interests or state ‘None declared’: None declared

Please leave your comments for the authors below

This study aimed to address an important research question (i.e. identifying the process of resilience building through group singing in people affected by cancer) through a grounded theory study.

Reviewing of relevant literature and description of the rationale for the study were done satisfactorily and the study resulted in a relatively strong theoretical model.

However, unfortunately the overall quality of the paper was diminished by several limitations. First, there was insufficient description on the grounded theory data analysis process (e.g., axial coding, selective coding, theoretical coding, memoing etc.). Second, there was a lack of clarity in describing and categorizing the sub-themes. There seemed to be overlap between some of the sub-themes and

the description of some sub-themes need more clarification or differentiation from the other themes. The connection between some of the inferences and the supporting quotes by participants was weak. Third, the description of the final grounded theory is also weak. Fourth, some of the inferences made in the discussion and conclusions sections are not justified by the results. Lastly, limitations of the study were not sufficiently addressed in the body of the paper.

Thus, I regret to inform the authors that the manuscript cannot be accepted with the current condition.

We'd like to thank Dr Shim for her comments and are pleased that she feels that the study addresses an important research question and that our analyses resulted in a relatively strong theoretical model. We also thank her for her feedback and address the issues raised as follows:

We apologise for the insufficient description of our analysis procedure. We have now explained in detail that we followed recommendation from Charmaz (2006, 2014), carrying out line-by-line initial coding, conceptual focused coding, axial coding to compare categories and sub-categories with one another and theoretical coding to explore and integrate these relationships, as well as creating memos to record emergent ideas.

We also apologise for the lack of clarity when describing and categorising sub-themes. We have now amended to label all of our sub-themes within the text to clearly show the process that we took to develop the themes described. We hope that this also shows how these themes are interrelated but distinct, addressing Dr Shim's comment here that there was overlap. In relation to the latter, we have now also included additional quotes to support the subthemes to highlight how they differentiate as well as explained a selection of quotes in greater detail to support our inferences.

We thank Dr Shim for recognising a need to describe our model in greater detail. We have now described the process for creating this model, explaining how diagramming, comparing and integrating themes resulted in a theory of perceived resilience. Further supporting this and responding to our first reviewer's comments regarding the data that supported the creation of each element of this model, we have now also included quotes within the diagram to support each component of the model with raw data.

We also apologise for not including limitations in the body of the paper and have now referenced limitations before our conclusion alongside thoughts for future avenues.

VERSION 2 – REVIEW

REVIEWER	Kathy Helzlsouer National Cancer Institute, U.S.A
REVIEW RETURNED	04-Sep-2018
GENERAL COMMENTS	The authors have responded adequately to the prior critique. Minor issues with tense agreement (e.g. "data" - plural)